# Physical fitness as a predictor of reaction time in soccer-playing children

Vanessa Santos[1, 2, ¤a, ¤b, ☺], Nuno Casanova[1, ¤a, ‡], Priscila Marconcin[1, 3, ¤a, ¤c, ‡], Renata Willig[1, ¤a, ‡], Josep Vidal-Conti[4, 5, ¤d, ¤e, ‡], Denise Soares[6, *, ¤f, ‡], Fábio Flôres[7, 8, 9, ¤g, ¤h]

**1** Insight: Piaget Research Center for Ecological Human Development, Piaget Institute, Almada, Portugal, **2** Faculty of Human Kinetics, Exercise and Health Laboratory, CIPER, University of Lisbon, 1495-751 Cruz Quebrada, Lisbon, Portugal, **3** Faculty of Health Sciences, Universidad Autónoma de Chile, Providencia, Chile, **4** Physical Activity and Sport Sciences Research Group (GICAFE), University of the Balearic Islands, Palma, Spain, **5** Institute for Educational Research and Innovation (IRIE), University of the Balearic Islands, Palma, Spain, **6** Liberal Arts Department, American University of the Middle East, Egaila, Kuwait, **7** Centro de Investigação em Educação e Psicologia (CIEP), Universidade de Évora, Évora, Portugal, **8** Comprehensive Health Research Centre (CHRC), Universidade de Évora, Évora, Portugal **9** Escola de Ciências Sociais, Universidade de Évora, Évora, Portugal

☺ These authors contributed equally to this work.
‡ NC, PM, RW, JVC, DS also contributed equally to this work.
¤aCurrent Address: Piaget Institute, Almada Campus, Almada, Portugal
¤bCurrent Address: Faculdade de Motricidade Humana, Universidade de Lisboa
¤cCurrent Address: Universidad Autónoma de Chile, Providencia, Chile
¤dCurrent Address: University of the Balearic Islands, Spain
¤eCurrent Address: University of the Balearic Islands, Spain
¤fCurrent Address: American University of the Middle East, Egaila, Kuwiat
¤gCurrent Address: Escola de Ciências Sociais, Universidade de Évora, Évora, Portugal
¤hCurrent Address: Escola de Saúde e Desenvolvimento Humano, Universidade de Évora, Évora, Portugal
* denise.soares@aum.edu.kw

## Abstract

This study investigated the relationship between reaction time (RT) and physical fitness (PF) in soccer-playing children, focusing on core strength, agility, flexibility and power. A sample of 89 boys (8.7 ± 2.2 years) participated in this investigation were conveniently chosen in a Portuguese soccer team. All participants were players in a non-competitive level. Data on PF components were collected and analysed to explore their associations with RT. The results revealed negative correlations between abdominal strength, agility, and RT, suggesting that stronger core stability and higher agility contribute to faster RT. However, other components, such as upper limb strength, flexibility, and horizontal jump performance, showed no significant correlation with RT. These findings highlight the importance of focusing on core strength and agility in training programs to enhance RT and overall performance in youth soccer. The study underscores the need for age-appropriate training interventions promoting physical and cognitive development.

## Introduction

Throughout childhood, various environments impact children's development, such as their home, community, parental work environments, friends' homes, schools, cultural influences, and sports participation [1]. These environments play a critical role in shaping motor skills

**Data availability statement:** You can find the data used for this research by using the DOI https://doi.org/10.5281/zenodo.14759924.

**Funding:** The author(s) received no specific funding for this work.

and physical fitness (PF), providing essential opportunities for children to engage in physical activity that promotes motor behavior [2]. Research has demonstrated that favorable environments that offer more opportunities can improve PF and the development of motor competence during childhood [3,4]. In this context, physical activities in such environments enhance motor skills and contribute to overall cognitive and social development during these developing years [2,5,6]. Favorable contexts can encourage and facilitate adjustments towards progressively more complex interactional activities in the immediate settings [7,8].

The sports environment, understood as one of the main contexts that children are present daily [1], provides participants of all ages and sex with many movement opportunities. Tailored training programs, specialized equipment, events, and interactions with fellow colleagues foster diverse possibilities for action and growth. Participation in sports can influence children physical literacy [9], aligning with principles of the Long-Term Athlete Development (LTAD) model, which emphasizes the importance of developing fundamental movement skills during critical developmental periods [10]. This approach not only supports physical literacy assumptions but also complements its inclusion as an important indicator in the internationally applied "Physical Activity Report Card" in several countries [11,12].

In this perspective, soccer is a demanding team sport, characterized by its intermittent nature, with oscillations between high-intensity activities, such as sprinting, jumping, and attacking, and lower-intensity phases, such as set-pieces and defensive movements [13,14]. This dynamic nature requires players to constantly adapt their physical capabilities, including strength, endurance, and reaction time (RT), to meet the demands of varying intensities during matches. This structure requires players to adapt constantly to the varying physiological requests of the game, particularly concerning their PF levels. Research shows that age and sex can influence soccer performance, affecting how players respond to these opportunities for action [15,16]. Additionally, coping with the game's tactical demands, such as switching between intense and less intense phases, is crucial for overall performance [17,18]. Therefore, in recent years, a growing body of evidence has evaluated these environments, with soccer representing one of the main areas of focus [19–21]. Despite that, few investigations to date tried to assess PF in soccer-playing boys [21–25]. This is a particularly notable gap given that young children experience significant physiological changes during childhood and adolescence, directly influencing their athletic performance [2,5,26]. This gap is particularly notable considering the importance of developing physical and motor attributes such as strength, endurance, RT (and response time), and agility during the formative years and development. These physical attributes are critical for soccer performance, as they enable participants to react quickly and efficiently to the fast-paced and dynamic nature of the sport [22,27,28]. As young participants develop, improving their motor skills and PF becomes crucial for long-term success and overall health [21,24,25]. For example, Reigal et al.[29] found that simple RT was related with children PF. Malina and colleagues [30] also showed higher levels of PF in children engaged in sports contexts.

Research has shown that sports are associated with improvements in RT [29]. In addition, sports promote a variety of movements that support its development [27,28,31]. For example, in other competitive or team sports, such as tennis and basketball, RT is crucial in numerous game situations where players must make quick decisions to succeed [27,32–34]. Additionally, studies have demonstrated that individuals with higher PF levels tend to have faster RT across various tasks [35,36]. Furthermore, information processing, cognition, and RT are indicators of decision-making speed and effectiveness in sports, which can significantly impact performance [37,38], even in soccer tasks [27,39]. Concerning soccer, the literature also showed that RT is associated with players balance [27], despite being choice [40], disjunctive [41], or simple RT [42]. Nevertheless, most of the literature so far usually

employ non-ecological assessments to evaluate RT in sport-related tasks to explore experienced players instead of young soccer players [43,44]. Also, research has shown that RT generally decreases from childhood through youth (improved performance), contributing to enhanced performance over time [39,45]. Those previous investigations have started to explore these aspects. Still, more targeted research is needed to establish a comprehensive profile of PF development in youth soccer and its influence on RT. Such data could provide valuable insights for coaches and sports scientists aiming to design age-appropriate training interventions that enhance performance while safeguarding the long-term health of young soccer players.

Few investigations have explored the associations between PF and RT in soccer-playing boys [21,25]. Therefore, this investigation aims to analyze the association between PF components (such as core strength, agility, flexibility and power) and RT in youth soccer players and evaluate the predictive capacity of these components on RT performance. By identifying the PF aspects most strongly associated with and predictive of RT, this investigation might provide valuable insights for optimizing training programs and fostering development in young soccer players. Thus, PF is expected to influence RT positively among soccer-playing boys.

## Materials and methods

### Sample

This investigation followed a cross-sectional study design. The sample size was determined using the G * Power v 3.1.9.7 software (Kiel University, Kiel, Germany) [46], considering Cohen's effect size of 0.35 for correlation bivariate normal model, error probability $\alpha = 0.05$, and $\beta = 0.95$. This calculation resulted in a sample size of 79 participants. The inclusion criteria included players between 6 to 12 years old without physical limitations or injuries. Participants with any developmental condition or who did not follow the procedures correctly were excluded from the final sample. Therefore, eighty-nine soccer-playing boys (mean age = 8.7 ± 2.2 years) were randomly recruited from a Portuguese soccer team to participate in the investigation (Table 1).

Table 1. Descriptive values of the sample.

| Variables | | Minimum | Maximum | M | SD |
|---|---|---|---|---|---|
| Characterization | Age (yrs) | 6.0 | 12.0 | 8.7 | 2.2 |
| | Weight (kg) | 18.3 | 59.7 | 31.5 | 9.7 |
| | Height (m) | 1.1 | 1.7 | 1.4 | 0.1 |
| | BMI (kg/m²) | 12.4 | 24.6 | 16.6 | 71.0 |
| Physical fitness | Upper limb strength (reps) | 1.0 | 45.0 | 11.0 | 7.7 |
| | Abdominal strength (reps) | 1.0 | 80.0 | 25.1 | 19.3 |
| | Agility (s) | 9.8 | 20.5 | 14.1 | 2.5 |
| | Upper limbs flexibility (cm) | 0.0 | 1.0 | 0.8 | 0.4 |
| | Lower limbs flexibility (cm) | 0.0 | 33.0 | 22.1 | 4.6 |
| | Horizontal jump (m) | 0.7 | 1.9 | 1.5 | 0.3 |
| | Shuttle (levels) | 1.0 | 8.0 | 3.6 | 1.7 |
| | Shuttle (routes) | 5.0 | 62.0 | 25.7 | 15.3 |
| Response time | Response time (ms) | 1094.0 | 3284.0 | 1769.6 | 462.5 |
| | Number of taps | 6.0 | 13.0 | 9.5 | 1.5 |

Legend: BMI – body mass index.

## Instruments and procedures

Oral consent was obtained from all participants and written consent from their parents/ guardians previously test beginning. The Instituto Piaget Ethics Committee approved the research (P12-S21-21.06.22), and the study followed all the Declaration of Helsinki guidelines [47]. Additional information regarding the ethical, cultural, and scientific considerations specific to inclusivity in global research is included in the Supporting Information.

Participants were briefed on the procedures, and their height was measured during inhalation using a stadiometer without wearing shoes (SECA 213, Bacelar & Irmão Lda, Portugal). Body weight was measured using a digital scale, with participants standing barefoot (SECA 761, Bacelar & Irmão Lda, Portugal). The recorded measurements were then used to calculate Body Mass Index (BMI) using the formula: $BMI = Weight (kg)/height (m)^2$. All assessments were conducted by an experienced researcher skilled in anthropometric measurements. Measurements were taken in the evening, at least three hours after the participants' last meal, with participants wearing their usual training equipment, excluding soccer boots.

The assessments were conducted over two training days, with a two-day interval between them. Players were assessed for their RT on the first day of evaluations, while on the second day, their PF was measured. Both assessments were conducted in a controlled environment, specifically a quiet room, before the players' soccer training sessions at 6 PM. This consistent setting ensured that external factors, such as noise or distractions, did not interfere with the results. Additionally, before each assessment, players completed a standardized warm-up consisting of jogging and stretching to ensure optimal physical readiness and reduce the risk of performance variability. The warm-up took 10 minutes.

## Reaction Time (response time) task

The task required the arrangement of four pods on the floor in a square formation, with 2 meters between each pod (BlazePod™, Tel Aviv, Israel)(see Fig 1). Players were instructed to

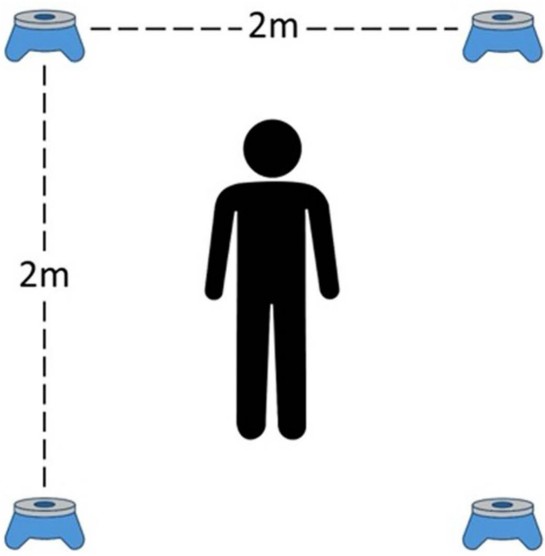

**Fig 1. Reaction time task setting.**

assume a position in the square's center and move within the square to contact the illuminated pod (indicated by a red light that turned on in a random order, with a time interval of 1.0 seconds to 1.5 seconds). Players were then required to complete the task as quickly as possible.

Three practice trials were conducted, comprising one trial for familiarization and two test trials. Each trial lasted 25 seconds, with a 15-second rest interval between trials. Then, the most successful attempt was employed for data analysis. While they received motivational feedback, no specific task results were provided.

### Physical fitness tests

The PF tests used in this study are based on the FitEscola battery, a validated Portuguese assessment tool designed to evaluate physical fitness in school-aged children and adolescents [48,49]. This battery includes a series of tests to measure key fitness components, such as strength, flexibility, agility, and aerobic capacity, emphasizing their relationship with health outcomes. The FitEscola protocol provides standardized procedures [48], ensuring the reliability and consistency of the data collected in youth populations.

**Upper Limb Strength (Push-ups Test).** Upper limb strength was assessed using the push-up test, which measures the endurance and strength of the upper body, particularly the chest, shoulders, and triceps. Participants were instructed to assume a prone position, with hands placed shoulder-width apart and feet together. They lowered their bodies until their elbows reached a 90-degree angle and then returned to the starting position by fully extending their arms. The goal was to complete as many push-ups as possible in 30 seconds, with the total number of correct repetitions recorded.

**Abdominal Strength (Sit-ups Test).** Abdominal strength was measured using the sit-ups test, which assesses the endurance of the abdominal muscles. Participants lay on their backs with knees bent at 90 degrees, feet flat on the floor, and arms crossed over the chest. They were instructed to perform as many sit-ups as possible in 30 seconds, with the correct form requiring the elbows to touch the knees on each repetition. The total number of correct sit-ups completed was recorded for analysis.

**Agility (4x10 Meter Shuttle Run Test).** The 4x10 meter shuttle run test assesses agility by combining maximum speed with coordination. Participants sprint between two lines 10 meters apart, retrieving and exchanging sponges placed at predetermined locations. The test evaluates acceleration, movement coordination, and execution speed. Starting from behind the initial line, the participant runs to the opposite line, picks up a sponge, returns to the start, exchanges it for another, and repeats the process for four sprints. The time is recorded using a stopwatch, with the best two attempts used for analysis.

**Upper Limb Flexibility (Back Scratch Test).** Upper limb flexibility was assessed using the back scratch test, which measures shoulder and upper arm flexibility. Participants reached one arm over the shoulder and the other behind the back, attempting to touch or overlap their fingers. The distance between the fingertips (if they did not meet) or the amount of overlap (if they did) was measured. The test was performed on both sides, and the best score was recorded.

**Lower Limb Flexibility (Sit-and-Reach Test).** Lower limb flexibility was evaluated using the sit-and-reach test, which measures the flexibility of the lower back and hamstrings. The participant sat on the floor with legs extended and feet flat against a sit-and-reach box. They reached forward with both hands, sliding them along the measurement scale as far as possible. The furthest point reached was recorded. The best score from two attempts was used for analysis.

**Horizontal Jump (Standing Long Jump Test).** The standing long jump test was used to measure lower body power. Participants stood behind a starting line with feet shoulder-width apart. They jumped forward as far as possible using a two-footed take-off, aiming for

maximum horizontal distance. The distance from the starting line to where the participant landed with their heels was measured. The best of two attempts was recorded for analysis.

**Shuttle Run (20-Meter Shuttle Run Test).** The 20-meter shuttle run (beep test) assessed aerobic endurance. Participants ran between two lines set 20 meters apart in time with beeps that gradually increased speed. The test continued until the participant could no longer maintain the pace. The highest level completed before dropping out was recorded as their score.

### Data Analysis

Descriptive analysis with mean and standard deviation, minimum and maximum values was used to characterize data. The Kolmogorov–Smirnov test confirms the data normality; therefore, the Pearson correlation was used to analyze the association between PF and RT. Correlation coefficients $< 0.30$ were considered weak, those between 0.30 and 0.70 were considered moderate, and coefficients $> 0.70$ were considered strong [50]. The multicollinearity analysis was performed to perform the multiple linear regression, and the association between the RT, as a dependent variable, and the model's independent variables (PF variables) was verified through the Nagelkerke $R^2$ (adjusted). The Statistical Package for Social Sciences (SPSS; IBM corporation), version 29.0, was used, adopting an alpha significance level of 5%.

### Results

Bivariate correlations examined the association between PF variables and RT in youth soccer players (Table 2). The results revealed moderate negative correlations between abdominal strength and RT, indicating that higher force production was associated with shorter RT. Additionally, agility demonstrated moderate negative correlations with RT and the number of taps, suggesting that participants with better agility exhibited improved performance in these areas. Despite these findings, age-related factors such as height and weight appear to indirectly influence RT, given their moderate to strong correlations with age and the developmental trajectory of neuromuscular coordination and muscle mass. These age-related effects may partly explain why younger participants exhibited lower performance in the RT task, as strength parameters develop progressively with age.

The $R^2 = 0.377$ showed that 37.7% of the variability in the agility component is explained by the RT indicating a moderate linear relationship, suggesting that other factors, are influencing this PF variable (Fig 2, a). Fig 2 (b) showed the associations between RT and shuttle measured by routes ($R^2 = 0.185$) indicating that only 18.5% of the variability in can be explained by RT through a weak linear relationship. Finally, Fig 2 (c) showed an extremely strong relationship between RT and the number of taps ($R^2 = 0.887$).

The model summary showed that the regression model explains approximately 88.6% of the variance in RT (adjusted $R^2 = 0.886$). This suggests that the independent variables included in the model provide a strong explanation for variations in response time. The multiple linear regression analysis results are presented in Tables 3 and 4. Table 3 provides the ANOVA results, indicating that the regression model is statistically significant ($p < 0.001$). Therefore, at least one of the predictor variables significantly influences RT.

Table 4 exhibits the regression coefficients for each independent variable. The most influential predictor was the number of taps, which significantly negatively affected response time ($B = -300.002$, $p < 0.001$). This indicates that the higher the number of taps, the lower the RT, which improves player performance. Additionally, agility was a significant predictor ($B = -23.681$, $p = 0.001$), showing that longer agility times result in increased RT. Other variables, such as Weight, height, BMI, upper body strength, and abdominal strength, did not show statistical significance.

**Table 2. Associations between PF variables and RT.**

| | Age | Weight | Height | BMI | Upper limb strength | Abdominal strength | Agil-ity | Upper limbs flexibility | Lower limbs flexibility | Horizon-tal jump | Shuttle (levels) | Shuttle (routes) | Response time | Number of taps |
|---|---|---|---|---|---|---|---|---|---|---|---|---|---|---|
| Age | 1 | .781* | .896* | −.118 | −.046 | .667* | −.576* | .207 | −.031 | .497* | .681* | .711* | −.633* | .716* |
| Weight | – | 1 | .877* | .760* | −.197 | .535* | −.367* | .162 | −.102 | .246* | .427* | .445* | −.565* | .612** |
| Height | – | – | 1 | −.101 | −.150 | .650* | −.554* | .173 | −.058 | .416* | .543* | .593* | −.707* | .774 |
| BMI | – | – | | 1 | −.088 | −.055 | .010 | .058 | −.052 | −.055 | −.099 | −.085 | .059 | −.106 |
| Upper limb strength | – | – | – | – | 1 | .162 | −.062 | −.044 | .035 | .128 | .145 | .187 | .063 | −.069 |
| Abdominal strength | – | – | – | – | – | | −.564* | .088 | .000 | .494* | .509* | .563** | −.583* | .648* |
| Agility | – | – | – | – | – | – | 1 | −.081 | −.181 | −.557* | −.455* | −.489** | −.614* | −.615* |
| Upper limbs flexibility | – | – | – | – | – | – | | 1 | .154 | .071 | .120 | .164 | −.103 | .143 |
| Lower limbs flexibility | – | – | – | – | – | – | – | – | 1 | .036 | .076 | .094 | .042 | −.006 |
| Horizontal jump | – | – | – | – | – | – | – | – | – | 1 | .549* | .559* | −.300* | .356* |
| Shuttle (levels) | – | – | – | – | – | – | – | – | – | – | 1 | .967* | −.378* | .465* |
| Shuttle (routes) | – | – | – | – | – | – | – | – | – | – | – | 1 | −.430* | .513* |
| Response time | – | – | – | – | – | – | – | – | – | – | – | – | 1 | −.942* |
| Number of taps | – | – | – | – | – | – | – | – | – | – | – | – | – | 1 |

Legend: BMI – body mass index

Note:

*$p < 0.05$

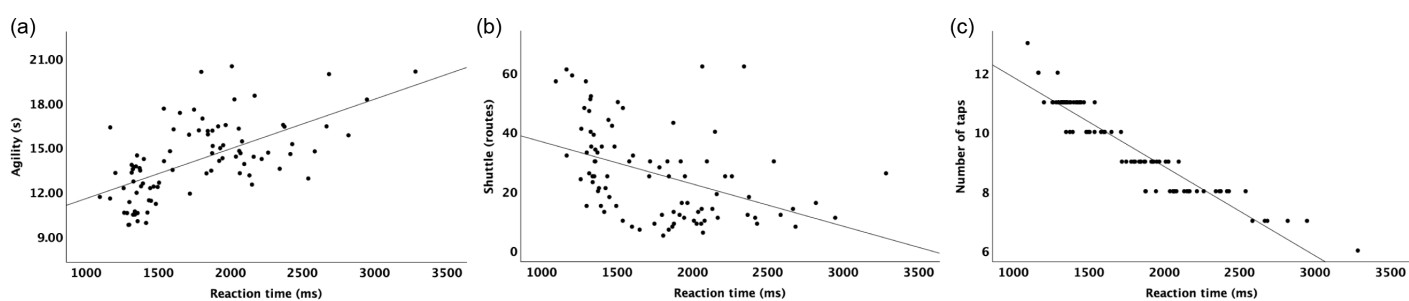

**Fig 2. Main associations concerning RT.**

## Discussion

This investigation aims to analyze the association between PF components and RT in youth soccer players, as well as to assess the predictive capacity of these components on RT performance, offering insights into the roles of strength and agility in enhancing performance. Our results demonstrate significant associations between specific PF components, namely abdominal strength, agility, and the RT, partially confirming our initial hypothesis that higher levels

of PF may be associated with shorter RT. Abdominal strength was significantly correlated with RT, with higher abdominal strength associated with shorter RT. This finding aligns with previous research highlighting the importance of core strength in stabilizing the body during rapid movements required in soccer [51,52]. A stronger core may allow for better postural control and quicker adaptations to sudden directional changes, which is essential for reacting promptly during matches. Maintaining balance and stability during high-intensity actions may enable players to respond more efficiently, supporting the notion that core strength is crucial for sports performance [53].

Agility also emerged as a significant predictor of RT, supporting the view that agility, the ability to change direction quickly while maintaining control, is essential for sports that demand rapid responses to dynamic situations [22]. In soccer, players constantly respond to stimuli such as the ball's movement or opponents' actions, making agility a vital component for maintaining performance [14]. The findings suggest that training programs to improve agility may enhance RT and improve soccer performance. However, these findings should be interpreted cautiously, as the influence of age-related factors, such as height and weight, may mediate these relationships. It may not be accurate to conclude that PF components directly improve RT, as other environmental and developmental factors likely play a significant role

**Table 3. ANOVA results.**

| Model | Sum of Squares | df | Mean Square | F | Sig. |
|---|---|---|---|---|---|
| Regression | 16836416.9 | 13 | 1295108.9 | 52.4 | 0.0[b] |
| Residual | 1805500.5 | 73 | 24732.9 | | |
| Total | 18641917.4 | 86 | | | |

[a]Dependent Variable: RT (ms)

[b]Predictors: (Constant), Number of taps, Lower limbs flexibility, Upper limb strength, Upper limbs flexibility, BMI (kg/m$^2$), Horizontal jump (m), Shuttle (levels), Agility (s), Abdominal strength, Age group, Height (m), Shuttle (routes), Weight (kg)

**Table 4. Coefficients of the multiple linear regression.**

| Model | Unstandardized Coefficients | | Standardized Coefficients | t | p | 95% Confidence Interval | | |
|---|---|---|---|---|---|---|---|---|
| | B | Std. Error | Beta | | | Lower Bound | Upper Bound | |
| (Constant) | 3232.2 | 1320.8 | | 2.45 | 0.02 | 599.93 | 5864.51 | |
| Age group | 26.6 | 52.8 | 0.0 | 0.50 | 0.62 | −78.62 | 131.74 | |
| Weight (kg) | −10.9 | 16.5 | −0.2 | −0.66 | 0.51 | −43.83 | 22.02 | |
| Height (m) | 581.1 | 867.0 | 0.2 | 0.67 | 0.51 | −1146.80 | 2309.09 | |
| BMI (kg/m$^2$) | 16.3 | 35.9 | 0.1 | 0.45 | 0.65 | −55.35 | 87.87 | |
| Upper limb strength | −0.5 | 2.5 | 0.0 | −0.21 | 0.83 | −5.58 | 4.49 | |
| Abdominal strength | 1.0 | 1.4 | 0.0 | 0.70 | 0.49 | −1.79 | 3.73 | |
| Agility (s) | 23.7 | 10.3 | 0.1 | 2.29 | 0.03 | 3.10 | 44.26 | |
| Upper limbs flexibility | 19.3 | 44.9 | 0.0 | 0.43 | 0.67 | −70.27 | 108.80 | |
| Lower limbs flexibility | 4.9 | 4.1 | 0.0 | 1.20 | 0.23 | −3.24 | 13.10 | |
| Horizontal jump (m) | 61.9 | 93.0 | 0.0 | 0.67 | 0.51 | −123.49 | 247.31 | |
| Shuttle (levels) | 39.3 | 43.6 | 0.1 | 0.90 | 0.37 | −47.54 | 126.07 | |
| Shuttle (routes) | −3.3 | 5.1 | −0.1 | −0.64 | 0.52 | −13.47 | 6.91 | |
| Number of taps | −300.0 | 20.6 | −1.0 | −14.58 | 0.00 | −341.00 | −259.00 | |

Legend: BMI – body mass index

a. Dependent Variable: RT (ms)

in shaping these associations. These findings align with previous studies highlighting the importance of agility and core strength for improving RT in youth. For example, the literature [51,52] has demonstrated that core stability supports fast directional changes and postural control, which are critical for RT in dynamic sports like soccer. Similarly, Reigal and colleagues [29] found that agility training enhances both physical and cognitive components of RT. However, our results differ from those of older or elite-level athletes, where strength or anaerobic power tends to have a greater impact [54]. These differences likely stem from variations in training levels, age, and developmental stages, emphasizing the need for age-specific interpretations.

Contrary to expectations, upper limb strength, flexibility (both upper and lower limbs), and horizontal jump performance were not significantly correlated with RT. While these fitness components contribute to overall physical performance, they may not directly influence RT in soccer. This aligns with previous findings indicating that upper limb strength, necessary in sports like swimming or weightlifting, has less of an impact on lower body-dominant sports like soccer [54]. Although beneficial for preventing injuries and increasing range of motion, flexibility may not be a critical factor in determining RT [4,55]. The use of a single RT assessment may not fully capture the extent of RT as a construct [56]. The addition of other RT assessments, such as stationary visual RT [27] could help to understand better the relationships with PF components in our sample. Furthermore, the homogeneity of our sample (youth soccer players from a single non-competitive team) may limit the generalizability of these findings to other populations or competitive levels. This methodological bias reinforces the need for caution when extrapolating these results to broader contexts.

These results have important implications for designing training programs for youth soccer players. Specifically, the findings highlight the need to focus on core strength and agility to improve RT and overall performance. Incorporating exercises that target the abdominal muscles, such as planks, leg raises, and sit-ups, may enhance core stability, which is crucial for balance and rapid movements [57]. Additionally, agility drills like shuttle runs, cone, and ladder drills may boost a player's ability to change direction quickly, improving RT and game performance [13]. The findings also underscore the importance of age-appropriate training interventions, particularly during the formative years of athletic development. As children undergo significant physiological changes during childhood and adolescence, designing training programs that enhance PF can preserve long-term health. Coaches and sports scientists must consider each player's developmental stage when creating fitness programs, focusing on improving fundamental movement skills and gradually introducing more sport-specific exercises as the player mature [10,24]. As also highlighted by the LTAD model, aligning training with biological growth and developmental phases is essential to optimize athletic potential, leveraging "windows of opportunity" for motor skill improvements. Therefore, promoting agility through physical education programs and youth sports may yield long-term benefits for athletic performance and overall health [58]. In addition, the strong correlation between agility and RT observed in this study suggests that agility training may also enhance cognitive function, as the ability to process and react to stimuli is closely linked to PF [59]. Incorporating exercises that challenge both the body and the mind, such as dual-task exercises or complex motor tasks, may be an effective strategy to improve both d cognitive function in soccer-playing boys [60].

Furthermore, literature has consistently shown that supportive and stimulating environments, which provide opportunities for diverse physical activities, are crucial in enhancing PF and fostering the development and competence in children [3,4]. Such environments promote improvements in fundamental motor skills and significantly

contribute to cognitive, emotional, and social development during early and middle childhood [61]. According to ecological theories [1,8], favorable contexts motivate children to explore and engage in progressively complex physical interactions, which help refine motor skills and adaptive behaviors [7,8]. Moreover, these enriched environments, such as sports contexts and, in our case, soccer training, we can assume that the children participation can potentially improve their PF and RT levels. This evidence underscores the importance of creating accessible and stimulating settings that allow children to explore, play, and interact actively.

Despite the significant findings, this investigation has some limitations. One key limitation is the composition of the sample, which consisted exclusively of soccer players from one Portuguese team with two years of experience at a non-competitive level. This sample homogeneity may restrict the generalizability of the findings to broader youth populations or players at different performance levels. Future research should aim to replicate these results with larger, more diverse, and competitively varied samples to enhance the applicability of our findings. Additionally, the cross-sectional nature of the research may prevent establishing causality between PF and RT. Longitudinal studies are needed to determine whether improvements in specific fitness components directly lead to enhancements in RT over time. Furthermore, our investigation lacked control over additional factors that might influence the observed associations, such as physical activity habits during leisure time. Investigating other relevant factors, including cognitive function, decision-making skills, and visual processing speed, may be particularly valuable in sports like soccer, where players must make complex decisions and react swiftly based on visual and spatial information. Another limitation relies in the design of the RT assessment, which involves movement over a 2-meter distance. Factors such as limb length, agility, and individual movement mechanics may influence the results, potentially confounding the assessment of RT. These biomechanical variations could introduce additional variability into the findings, making it challenging to isolate RT as a standalone measure. Future studies should consider adjusting test protocols or incorporating biomechanical analyses to account for these influences and provide a more precise evaluation of RT in this population.

## Practical application

Our investigation provides practical insights for coaches and practitioners seeking to improve performance in youth soccer players. Core strength and agility have emerged as predictors of RT, underscoring the importance of targeted training exercises to develop these components of PF. However, it is important to recognize that physiological differences between a non-competitive sample and youth-level soccer players may influence the effectiveness of such interventions. To improve RT in youth players, coaches should focus on building core stability through progressive and age-appropriate exercises, such as modified planks, leg raises, and dynamic core movements. Strengthening the core improves body control and balance, allowing faster and more efficient movements suited to the demands of soccer. For agility development, training sessions should include drills such as shuttle runs, cone weaving, and ladder exercises tailored to the physical capabilities of younger players. Emphasizing proper movement mechanics during these drills is crucial to optimize performance enhancements. This includes focusing on fundamental movement skills in the early stages and gradually introducing sport-specific drills as players mature. Finally, incorporating activities that stimulate cognitive processes, such as decision-making and visual processing, can help improve RT and mimic the complex demands of in-game scenarios. Such an approach might ensure the long-term development of youth players, preparing them to transition effectively to higher levels and competition.

## Conclusion

This study demonstrated the significant roles of core strength and agility in improving RT among youth soccer players. The findings highlight that abdominal strength and agility are essential for faster RT, while other factors, such as upper limb strength, flexibility, and horizontal jump performance, appear less influential in this context. Prioritizing core strength and agility in training programs could enhance soccer-playing boys physical and reaction capacities, enabling them to perform more efficiently during high-intensity actions. These results reinforce the importance of incorporating targeted exercises that develop these specific components to optimize sports performance in soccer.

## Supporting information

**S1 File. Inclusivity-in-global-research-questionnaire.**
(DOCX)

## Acknowledgments

The authors would like to express their sincere gratitude to Gonçalo Rodrigues, Tiago Martins, Paulo Reis, Pedro Moga, and Pedro Ambrósio, third-year students of the Physical Education and Sports degree program at Institute Piaget of Almada, during the 2023/2024 academic year, for their invaluable assistance in collecting field data. Their dedication and contributions were essential to the successful completion of this study.

## Author contributions

**Conceptualization:** Vanessa Santos.

**Data curation:** Vanessa Santos, Nuno Casanova, Renata Willig, Fábio Flôres.

**Formal analysis:** Renata Willig, Josep Vidal-Conti.

**Funding acquisition:** Denise Soares.

**Investigation:** Priscila Marconcin.

**Methodology:** Nuno Casanova, Priscila Marconcin, Renata Willig.

**Software:** Josep Vidal-Conti.

**Supervision:** Fábio Flôres.

**Validation:** Priscila Marconcin, Josep Vidal-Conti, Denise Soares, Fábio Flôres.

**Visualization:** Priscila Marconcin.

**Writing – original draft:** Vanessa Santos, Nuno Casanova, Josep Vidal-Conti.

**Writing – review & editing:** Denise Soares, Fábio Flôres.

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
