## [Decision Letter · Decision Letter 0]

20 Jan 2025

PONE-D-24-54366Physical Fitness as a Predictor of Reaction Time in Youth Football PlayersPLOS ONE

Dear Dr. Paschoal Soares,

Thank you for submitting your manuscript to PLOS ONE. After careful consideration, we feel that it has merit but does not fully meet PLOS ONE’s publication criteria as it currently stands. Therefore, we invite you to submit a revised version of the manuscript that addresses the points raised during the review process.

We look forward to receiving your revised manuscript.

Kind regards,

Mário Espada, PhD

Academic Editor

PLOS ONE

Journal Requirements:

2. Please include a complete copy of PLOS’ questionnaire on inclusivity in global research in your revised manuscript. Our policy for research in this area aims to improve transparency in the reporting of research performed outside of researchers’ own country or community. The policy applies to researchers who have travelled to a different country to conduct research, research with Indigenous populations or their lands, and research on cultural artefacts. The questionnaire can also be requested at the journal’s discretion for any other submissions, even if these conditions are not met.  Please find more information on the policy and a link to download a blank copy of the questionnaire here: https://journals.plos.org/plosone/s/best-practices-in-research-reporting. Please upload a completed version of your questionnaire as Supporting Information when you resubmit your manuscript

Reviewers' comments:

Reviewer's Responses to Questions

**Comments to the Author**

1. Is the manuscript technically sound, and do the data support the conclusions?

Reviewer #1: Yes

Reviewer #2: Yes

Reviewer #3: Yes

Reviewer #4: No

2. Has the statistical analysis been performed appropriately and rigorously? 

Reviewer #1: Yes

Reviewer #2: Yes

Reviewer #3: Yes

Reviewer #4: No

3. Have the authors made all data underlying the findings in their manuscript fully available?

Reviewer #1: Yes

Reviewer #2: Yes

Reviewer #3: Yes

Reviewer #4: Yes

4. Is the manuscript presented in an intelligible fashion and written in standard English?

Reviewer #1: Yes

Reviewer #2: Yes

Reviewer #3: Yes

Reviewer #4: Yes

5. Review Comments to the Author

Reviewer #1: Physical Fitness as a Predictor of Reaction Time in Youth Football Players

Or

A cross-sectional study of how environmental and school variables shape schoolchildren's motor competence

Title and Authors

- The order of authors and the title of the paper are not the same in the submission and in the document sent. Please revise.

- I would like to suggest using “children” instead of “youth football players” as the sample is composed by sports participants with 6 to 10 years of age. Also, if it was in soccer, I would suggest changing from “football” to “soccer” to be easily found in databases.

Abstract

- Line 28 – Please consider including “male” for a better clarity.

- Line 28 – considering the included variables, what the authors mean by “coordination”?

- Line 29 – if possible, include the PF analysed. Also, as indicated earlier, it should be also indicated coordination/motor competence here.

- I feel that is missing some information about coordination. It was decisive or not? Please consider adding some information or removing this from the aim.

Introduction

In general, I feel that this session could benefit for a clarity on the necessity of the present study, i.e., clarifying the importance of RT in soccer, and specially the capacity of children to develop speed in those ages.

- Line 53 to 55 – Consider changing to “Football is characterized by its intermittent nature, with oscillations between high-intensity activities, such as sprinting, jumping, and attacking, and lower-intensity phases, such as set-pieces and defensive movements” to have more coherence in the writing.

- Line 56 to 58 and lines 69 to 71 – so why the authors chose RT to be the main capacity to be explored? Could it be related to the age of the participants? Or it is related to the importance in football (which is?)?

- Line 66 to 69 - I would suggest using more the word “participants” rather than “athletes”, considering the age of the sample.

- Line 77 to 78 – I would suggest focussing in football as it is the main aim of the present study, and strengthened the need of this study in analysing RT in those ages with participants with this specific practice (football).

Material and Methods

- Line 103 – I would prefer “participants”. Please consider in the entire manuscript.

- Line 136 – please consider removing the first sentence.

Results

- This section seems clear and complete to me. I do not suggest any changing or any further information.

Discussion

- Line 264 – the term “coordination” appears again. I would like to ask the authors to clarify what do they mean by coordination.

- Line 277 – is missing a reference here.

- Line 282 – why “Contrary to expectations”?

- Line 294 – Please see the paper below and reconsider this sentence: https://www.thieme-connect.com/products/ejournals/abstract/10.1055/a-1157-9078

- Line 299 and 300 – I would suggest removing “and prevent overtraining and injuries is crucial” because the literature seems not so sure about this affirmation, especially in those ages.

- Line 300 to 302 – In my opinion, this idea should be explored in the manuscript, not only in the discussion session, but also in the introduction for a better contextualization of this scientific problem that the authors are analyzing. We know that the maturation and growth process could influence the “importance” of the different capacities in different years of the development of a youth. I would advise the authors to also see the Long Term Athlete Development (LTAD) model to strengthened the importance of analyzing RT at these ages.

- Line 306 to 310 – As the aim of the present study is sports and not health, I would like to suggest removing this idea.

- Line 312 to 313 – “mental coordination” or “working memory” or “executive functions”, what the authors really want to mean?

- Line 316 to 327 – I would suggest including (or incorporating, since the authors already have) this information in the introduction section. Nevertheless, considering that the aim of the present study is sports and not health and the authors did not conduct an intervention, I would like to suggest removing this paragraph.

- Line 329 – considering that the GPower mentioned 79 participants and the authors evaluated 89, I this that this is not a limitation.

Reviewer #2: Dear Authors

Thank you for this carefully crafted study. With a few minor revisions, I believe the manuscript has the potential to make a meaningful contribution to understanding the effects of physical fitness components on reaction time in youth football players.

You can find my recommendations below:

The materials and methods section should include information on the pre-test warm-up and nutrition, as both can have a substantial effect on the results

Could you elaborate on the study's limitations, especially in relation to the participants' performance levels? It would be useful to understand how these levels may have impacted on the results and the overall applicability of the findings.

Please explain how potential physiological differences between youth and elite level football players may affect these outcomes and provide potential solutions to overcome this issue.

Best wishes

Reviewer #3: I would like to appretiate the efforts of the authors in implementing the project and writing this article “Physical Fitness as a Predictor of Reaction Time in Youth Football Players”.

The aim of this study is just to analyze the association between physical fitness components (such as core strength, agility, flexibility, power, and coordination) and reaction time in youth football players and evaluate the predictive capacity of these components on youth reaction time performance. This approach will help identify the aspects of physical fitness that are most strongly associated with and can predict improvements in time, which is crucial for in-game among football players. While physical fitness is expected to influence reaction time positively among football players.

I have these comments and questions:

The work brings interesting results, which can be very useful and beneficial for the practice. However, I have two comments and questions.

Line 132: jogging during the warm-up lasted only 1 minute? Was it enough? How long did the stretching last?

Line 359: I'm not sure if any of the tests are aimed directly at core strength.

Reviewer #4: The manuscript addresses an interesting research question, examining the relationship between physical fitness (PF) components and reaction time (RT) in youth football players. However, significant methodological concerns, inadequate data interpretation, and limited scientific contribution raise serious concerns about the validity of the study.

Introduction

Contextualization and Theoretical Framework: The introduction provides a broad background but lacks a detailed theoretical framework linking specific physical fitness components to reaction time. Expanding on the mechanisms through which agility, core strength, and other fitness components influence RT would strengthen the rationale for the study.

Age-Related Variability: Given the wide age range of participants (5–14 years), the introduction should address how developmental differences might impact physical fitness and RT outcomes. A brief review of age-related changes in motor and cognitive functions relevant to RT could provide a clearer context for the research.

Methodology

Sample Selection and Age Range Justification: The inclusion of participants aged 5 to 14 years introduces significant variability due to differing developmental stages. Justification for this wide age range should be provided. Alternatively, a stratified analysis by age groups could offer more meaningful insights. "The age range of participants stated in the abstract (aged 6 to 10 years) differs from that specified in the methods section (players between 5 to 14 years old) and Table 1 (5 to 13 years)."

Reaction Time Test Protocol: The choice of specific reaction test as the RT test protocol raises concerns about methodological bias. Since the test involves movement over a 2-meter distance, factors such as limb length and agility may influence results beyond pure reaction time. If this test was selected for its relevance to football, further justification of its ecological validity is needed. Moreover, discussing the reliability of the chosen test in this context and considering a stationary RT test as an alternative would improve methodological robustness.

Single RT Test Limitation: Measuring RT using a single test limits the study’s ability to draw comprehensive conclusions. Different types of RT tests (e.g., stationary visual RT or auditory RT) could yield different relationships with physical fitness components. Including multiple RT assessments or acknowledging this limitation explicitly would enhance the study’s credibility.

Confounding Factors: The study does not adequately control for potential confounders such as previous training experience, nutritional status, or sleep quality, which could influence both physical fitness and RT. These should be discussed as limitations or adjusted for in the analysis.

Results

Age and Height Effects: The findings indicate that age and height have the strongest correlations with RT, suggesting that these factors may play a more significant role than the selected physical fitness parameters. This should be emphasized and discussed in detail, as it may affect the interpretation of the results.

Scatterplots for Key Correlations: Adding scatterplots for significant correlations (e.g., between agility and RT) would enhance the clarity of data presentation and help readers better understand the relationships.

Effect Sizes and Confidence Intervals: Reporting effect sizes and confidence intervals for key findings would improve the interpretation of the study’s practical significance.

Discussion

Generalization of Findings: The discussion should avoid overgeneralizing the findings. Given the methodological biases and the influence of age and height on RT, it may not be accurate to conclude that specific physical fitness components directly improve RT.

Comparison with Previous Studies: A more thorough comparison with existing literature on PF and RT in youth athletes is necessary. How do these findings align or contrast with previous research?

Limitations Section: A dedicated "Limitations" subsection should be added to discuss the following issues:

The use of a convenience sample, which may limit the generalizability of findings.

The broad age range, which introduces variability due to developmental differences.

The use of a single RT test protocol, potentially biasing results.

Potential unmeasured confounders, such as previous training, nutrition, and sleep.

Minor Revisions

Abstract

Limitations and Future Directions: The abstract should briefly mention key limitations (e.g., broad age range, single RT test protocol) and suggest directions for future research.

The age range of participants stated in the abstract (aged 6 to 10 years) differs from that specified in the methods section (players between 5 to 14 years old) and Table 1 (5 to 13 years).

Figures and Tables

Scatterplots: Consider adding scatterplots for key correlations to improve data visualization.

Confidence Intervals: Ensure that all tables include confidence intervals for regression coefficients to enhance interpretability.

References

Completeness: Ensure that all key references on PF and RT in youth populations are included.

Formatting: Verify that all references are correctly formatted according to journal guidelines.

6. PLOS authors have the option to publish the peer review history of their article (what does this mean? ). If published, this will include your full peer review and any attached files.

**Do you want your identity to be public for this peer review?** For information about this choice, including consent withdrawal, please see our Privacy Policy .

Reviewer #1: **Yes: ** Ana Filipa Braga Barroso Campos Silva

Reviewer #2: No

Reviewer #3: No

Reviewer #4: **Yes: ** Cihan Aygün

---

## [Author Response · Author response to Decision Letter 1]

2 Feb 2025

Dear Editor,

Thank you for the opportunity to improve the quality of our work. We followed all recommendations, accordingly. All answers were provided in red in this file and the manuscript for better clarity.

Reviewer #1: Physical Fitness as a Predictor of Reaction Time in Youth Football Players

Or A cross-sectional study of how environmental and school variables shape schoolchildren's motor competence

R: Thank you for your question. The title was changed to "Physical Fitness as a Predictor of Reaction Time in Soccer-Playing Children" was suggested.

- The order of authors and the title of the paper are not the same in the submission and in the document sent. Please revise.

R: Thank you for pointing out. Corrections were performed.

- I would like to suggest using "children" instead of "youth football players" as the sample is composed by sports participants with 6 to 10 years of age. Also, if it was in soccer, I would suggest changing from "football" to "soccer" to be easily found in databases.

R: Thank you for your question. The title was changed to "Physical Fitness as a Predictor of Reaction Time in Soccer-Playing Children" was suggested.

Abstract

- Line 28 – Please consider including "male" for a better clarity.

R: Thank you for the suggestion.

- Line 28 – considering the included variables, what the authors mean by "coordination"?

R: Thank you for the suggestion.

- Line 29 – if possible, include the PF analysed. Also, as indicated earlier, it should be also indicated coordination/motor competence here.

R: Thank you for the suggestion. The PF variables were mentioned in the abstract.

- I feel that is missing some information about coordination. It was decisive or not? Please consider adding some information or removing this from the aim.

R: Thank you. The coordination variable was misplaced and was removed accordingly.

Introduction

In general, I feel that this session could benefit for a clarity on the necessity of the present study, i.e., clarifying the importance of RT in soccer, and specially the capacity of children to develop speed in those ages.

R: Thank you. The introduction section was reviewed.

- Line 53 to 55 – Consider changing to "Football is characterized by its intermittent nature, with oscillations between high-intensity activities, such as sprinting, jumping, and attacking, and lower-intensity phases, such as set-pieces and defensive movements" to have more coherence in the writing.

R: Thank you for the suggestion. We performed the changes.

- Line 56 to 58 and lines 69 to 71 – so why the authors chose RT to be the main capacity to be explored? Could it be related to the age of the participants? Or it is related to the importance in football (which is?)?

R: Thank you. We strongly believe, and it is also pointed out in the literature, the importance of RT for soccer. The introduction section was reviewed.

- Line 66 to 69 - I would suggest using more the word "participants" rather than "athletes", considering the age of the sample.

R: Changes were performed accordingly.

- Line 77 to 78 – I would suggest focussing in football as it is the main aim of the present study, and strengthened the need of this study in analysing RT in those ages with participants with this specific practice (football).

R: Changes were performed accordingly.

Material and Methods

- Line 103 – I would prefer "participants". Please consider in the entire manuscript.

R: Changes were performed accordingly.

- Line 136 – please consider removing the first sentence.

R: Changes were performed accordingly.

Discussion

- Line 264 – the term "coordination" appears again. I would like to ask the authors to clarify what do they mean by coordination.

R: Changes were performed accordingly.

- Line 277 – is missing a reference here.

R: Thank you. The following reference supported our affirmation: "Strykalenko Y, Huzar V, Shalar O, Oloshynov S, Homenko V, Svirida V. Physical fitness assessment of young football players using an integrated approach. Journal of Physical Education and Sport. 2021 Jan 1;21(1):360–6.”

- Line 294 – Please see the paper below and reconsider this sentence: https://www.thieme-connect.com/products/ejournals/abstract/10.1055/a-1157-9078

R: Thank you.

- Line 299 and 300 – I would suggest removing "and prevent overtraining and injuries is crucial" because the literature seems not so sure about this affirmation, especially in those ages.

R: Thank you.

- Line 300 to 302 – In my opinion, this idea should be explored in the manuscript, not only in the discussion session, but also in the introduction for a better contextualization of this scientific problem that the authors are analyzing. We know that the maturation and growth process could influence the "importance" of the different capacities in different years of the development of a youth. I would advise the authors to also see the Long Term Athlete Development (LTAD) model to strengthened the importance of analyzing RT at these ages.

R: Thank you. Changes were performed through the text.

- Line 306 to 310 – As the aim of the present study is sports and not health, I would like to suggest removing this idea.

R: Thank you for the suggestion. The paragraph was removed.

- Line 312 to 313 – "mental coordination" or "working memory" or "executive functions", what the authors really want to mean?

R: Changes were performed.

- Line 316 to 327 – I would suggest including (or incorporating, since the authors already have) this information in the introduction section. Nevertheless, considering that the aim of the present study is sports and not health and the authors did not conduct an intervention, I would like to suggest removing this paragraph.

R: Changes were performed.

- Line 329 – considering that the GPower mentioned 79 participants and the authors evaluated 89, I this that this is not a limitation.

R: Changes were performed.

Reviewer #2:

The materials and methods section should include information on the pre-test warm-up and nutrition, as both can have a substantial effect on the results

R: Thank you for your suggestion. This information was already provided by the authors in the procedures section: "Measurements were taken in the evening, at least three hours after the participants' last meal, with participants wearing their usual training equipment, excluding soccer boots; and before each assessment, players completed a standardized warm-up consisting of one minute of jogging and stretching to ensure optimal physical readiness and reduce the risk of performance variability."

Could you elaborate on the study's limitations, especially in relation to the participants' performance levels? It would be useful to understand how these levels may have impacted on the results and the overall applicability of the findings.

R: Changes were performed in the discussion section.

Please explain how potential physiological differences between youth and elite level football players may affect these outcomes and provide potential solutions to overcome this issue.

R: Changes were performed in the discussion section.

Reviewer #3:

Line 132: jogging during the warm-up lasted only 1 minute? Was it enough? How long did the stretching last?

R: Thank you for noticing that. The whole warm-up consisted of 10 minutes in which participants jogged and stretched, following the team's planning and schedule.

Line 359: I'm not sure if any of the tests are aimed directly at core strength.

R: Thank you for raising this point. The push-ups and sit-ups tests employed in our study assess core strength, as they require activation and stabilization of core muscles during execution. The following reference supported our affirmation: Oliva-Lozano, J., & Muyor, J. (2020). Core muscle activity during physical fitness exercises: A systematic review. In International Journal of Environmental Research and Public Health (Vol. 17, Issue 12, pp. 1–42). MDPI AG. https://doi.org/10.3390/ijerph17124306

Reviewer #4:

Introduction

Contextualization and Theoretical Framework: The introduction provides a broad background but lacks a detailed theoretical framework linking specific physical fitness components to reaction time.

R: Thank you for the suggestion. The introduction was changed using the comments of all reviewers.

Expanding on the mechanisms through which agility, core strength, and other fitness components influence RT would strengthen the rationale for the study.

R: Thank you for the suggestion. The introduction was changed using all reviewers' comments to improve the rationale.

Age-Related Variability: Given the wide age range of participants (5–14 years), the introduction should address how developmental differences might impact physical fitness and RT outcomes. A brief review of age-related changes in motor and cognitive functions relevant to RT could provide a clearer context for the research.

R: Thank you for the suggestion. The introduction was changed using the comments of all reviewers. In addition, the age range was correct to 6 to 12 years.

Methodology

Sample Selection and Age Range Justification: The inclusion of participants aged 5 to 14 years introduces significant variability due to differing developmental stages. Justification for this wide age range should be provided. Alternatively, a stratified analysis by age groups could offer more meaningful insights. "The age range of participants stated in the abstract (aged 6 to 10 years) differs from that specified in the methods section (players between 5 to 14 years old) and Table 1 (5 to 13 years)."

R: Thank you for the suggestion. In fact, the age range was wrongly inserted, and the corrections were performed through the manuscript (correct is 6 to 12). Despite that, it wasn't one of our goals to compare different age ranges in this investigation, even though we believe that is an important issue to be addressed in our future investigation.

Reaction Time Test Protocol: The choice of specific reaction test as the RT test protocol raises concerns about methodological bias. Since the test involves movement over a 2-meter distance, factors such as limb length and agility may influence results beyond pure reaction time. If this test was selected for its relevance to football, further justification of its ecological validity is needed. Moreover, discussing the reliability of the chosen test in this context and considering a stationary RT test as an alternative would improve methodo-logical robustness.

R: Thank you for the important questions. In fact, those questions addressed the limitations during the discussion section.

Single RT Test Limitation: Measuring RT using a single test limits the study's ability to draw comprehensive conclusions. Different types of RT tests (e.g., stationary visual RT or auditory RT) could yield different relationships with physical fitness components. Including multiple RT assessments or acknowledging this limitation explicitly would enhance the study's credibility.

R: Thank you for the critical questions. Some of the suggestions were added to the manuscript. Despite that, we do not believe that auditory RT is important to be evaluated here due to the lack of ecological validity concerning soccer.

Confounding Factors: The study does not adequately control for potential confounders such as previous training experience, nutritional status, or sleep quality, which could influence both physical fitness and RT. These should be discussed as limitations or adjusted for in the analysis.

R: The limitations were improved.

Results

Age and Height Effects: The findings indicate that age and height have the strongest correlations with RT, suggesting that these factors may play a more significant role than the selected physical fitness parameters. This should be emphasized and discussed in detail, as it may affect the interpretation of the results.

R: Thank you for your valuable feedback. We have adjusted the manuscript to address your concerns regarding the influence of age, height, and weight on RT.

Scatterplots for Key Correlations: Adding scatterplots for significant correlations (e.g., between agility and RT) would enhance the clarity of data presentation and help readers better understand the relationships.

R: Thank you for the suggestion. Some of the main associations were presented in scatterplots.

Effect Sizes and Confidence Intervals: Reporting effect sizes and confidence intervals for key findings would improve the interpretation of the study's practical significance.

R: Thank you for the suggestion. The changes were made.

Discussion

Generalization of Findings: The discussion should avoid overgeneralizing the findings. Given the methodo-logical biases and the influence of age and height on RT, it may not be accurate to conclude that specific physical fitness components directly improve RT.

R: Thank you for your insightful observation. We have revised the discussion to address the potential for overgeneralization of the findings.

Comparison with Previous Studies: A more thorough comparison with existing literature on PF and RT in youth athletes is necessary. How do these findings align or contrast with previous research?

Limitations Section: A dedicated "Limitations" subsection should be added to discuss the following issues:

R: Thank you for your valuable feedback. We have adjusted both suggestions in the manuscript.

The use of a convenience sample, which may limit the generalizability of findings.

R: The limitations were improved.

The broad age range, which introduces variability due to developmental differences.

R: The limitations were improved.

The use of a single RT test protocol, potentially biasing results.

R: The limitations were improved.

Potential unmeasured confounders, such as previous training, nutrition, and sleep.

R: The limitations were improved.

Minor Revisions

Abstract

Limitations and Future Directions: The abstract should briefly mention key limitations (e.g., broad age range, single RT test protocol) and suggest directions for future research.

R: Changes were performed in the abstract.

The age range of participants stated in the abstract (aged 6 to 10 years) differs from that specified in the methods section (players between 5 to 14 years old) and Table 1 (5 to 13 years).

R: Changes were performed in the abstract.

Figures and Tables

Scatterplots: Consider adding scatterplots for key correlations to improve data visualization.

R: Thank you for the suggestion. Some of the main associations were presented in scatterplots.

Confidence Intervals: Ensure that all tables include confidence intervals for regression coefficients to enhance interpretability.

R: Thank you for the suggestion. The confidence intervals were added in Table 4.

References

Completeness: Ensure that all key references on PF and RT in youth populations are included.

Formatting: Verify that all references are correctly formatted according to journal guidelines.

R: A complete review of the references was performed.

---

## [Decision Letter · Decision Letter 1]

12 Feb 2025

Physical Fitness as a Predictor of Reaction Time in Youth Football Players

PONE-D-24-54366R1

Dear Dra. Denise Paschoal Soares,

We’re pleased to inform you that your manuscript has been judged scientifically suitable for publication and will be formally accepted for publication once it meets all outstanding technical requirements.

Kind regards,

Mário Espada

Academic Editor

PLOS ONE

Reviewers' comments:

Reviewer's Responses to Questions

**Comments to the Author**

1. If the authors have adequately addressed your comments raised in a previous round of review and you feel that this manuscript is now acceptable for publication, you may indicate that here to bypass the “Comments to the Author” section, enter your conflict of interest statement in the “Confidential to Editor” section, and submit your "Accept" recommendation.

Reviewer #1: All comments have been addressed

Reviewer #3: All comments have been addressed

2. Is the manuscript technically sound, and do the data support the conclusions?

Reviewer #1: Yes

Reviewer #3: Yes

3. Has the statistical analysis been performed appropriately and rigorously? 

Reviewer #1: Yes

Reviewer #3: (No Response)

4. Have the authors made all data underlying the findings in their manuscript fully available?

Reviewer #1: Yes

Reviewer #3: Yes

5. Is the manuscript presented in an intelligible fashion and written in standard English?

Reviewer #1: Yes

Reviewer #3: Yes

6. Review Comments to the Author

Reviewer #1: Dear Authors,

Thank you for considering my suggestions. I am satisfied with this new version. Congratulations on your work.

Reviewer #3: There have been considerable modifications to the article, I think the article is suitable for publication. My comments have been taken into account and explained. I have no further questions or comments.

7. PLOS authors have the option to publish the peer review history of their article (what does this mean? ). If published, this will include your full peer review and any attached files.

**Do you want your identity to be public for this peer review?** For information about this choice, including consent withdrawal, please see our Privacy Policy .

Reviewer #1: **Yes**

Reviewer #3: No

---

## [Editor Report · Acceptance letter]

PONE-D-24-54366R1

PLOS ONE

Dear Dr. Soares,

I'm pleased to inform you that your manuscript has been deemed suitable for publication in PLOS ONE. Congratulations! Your manuscript is now being handed over to our production team.

Kind regards,

on behalf of

Dr. Mário Espada

Academic Editor

PLOS ONE